# Park Proximity and Use for Physical Activity among Urban Residents: Associations with Mental Health

**DOI:** 10.3390/ijerph17134885

**Published:** 2020-07-07

**Authors:** Stephanie L. Orstad, Kristin Szuhany, Kosuke Tamura, Lorna E. Thorpe, Melanie Jay

**Affiliations:** 1Department of Medicine, Division of General Internal Medicine and Clinical Innovation, New York University (NYU) Grossman School of Medicine, New York, NY 10016, USA; 2Department of Psychiatry, NYU Grossman School of Medicine, New York, NY 10016, USA; kristin.szuhany@nyulangone.org; 3Social Determinants of Obesity and Cardiovascular Risk Laboratory, Cardiovascular Branch, Division of Intramural Research, National Heart, Lung and Blood Institute, National Institutes of Health, Bethesda, MD 20892, USA; kosuke.tamura@nih.gov; 4Department of Population Health, Division of Epidemiology, NYU Grossman School of Medicine, New York, NY 10016, USA; lorna.thorpe@nyulangone.org; 5Departments of Medicine and Population Health, NYU Grossman School of Medicine, New York, NY 10016, USA; melanie.jay@nyulangone.org; 6Comprehensive Program on Obesity, NYU Langone Health, New York, NY 10016, USA

**Keywords:** neighborhood, urban, greenspace, recreation, safety, crime, depression, quality of life

## Abstract

Increasing global urbanization limits interaction between people and natural environments, which may negatively impact population health and wellbeing. Urban residents who live near parks report better mental health. Physical activity (PA) reduces depression and improves quality of life. Despite PA’s protective effects on mental health, the added benefit of urban park use for PA is unclear. Thus, we examined whether park-based PA mediated associations between park proximity and mental distress among 3652 New York City residents (61.4% 45 + years, 58.9% female, 56.3% non-white) who completed the 2010–2011 Physical Activity and Transit (PAT) random-digit-dial survey. Measures included number of poor mental health days in the previous month (outcome), self-reported time to walk to the nearest park from home (exposure), and frequency of park use for sports, exercise or PA (mediator). We used multiple regression with bootstrap-generated 95% bias-corrected confidence intervals (BC CIs) to test for mediation by park-based PA and moderation by gender, dog ownership, PA with others, and perceived park crime. Park proximity was indirectly associated with fewer days of poor mental health via park-based PA, but only among those not concerned about park crime (index of moderated mediation = 0.04; SE = 0.02; 95% BC CI = 0.01, 0.10). Investment in park safety and park-based PA promotion in urban neighborhoods may help to maximize the mental health benefits of nearby parks.

## 1. Introduction

Today, the world population is increasingly concentrated in cities. In 2018, 82% of North Americans and 55% of people worldwide lived in urban areas [1]. Living in large cities limits regular contact with the natural environment and the associated physical and mental health benefits. Lack of quality greenspace may exacerbate downward trends in urban residents’ subjective wellbeing, particularly those of low-income, in the world’s most economically developed cities [2]. The prevalence of chronic diseases (e.g., obesity) and depression have increased significantly in the United States (USA) and globally in recent decades [3,4,5]. In New York City (NYC), the prevalence of obesity has increased to 32.4% and the prevalence of depression has remained steady at approximately 9% [6,7]. In 2014, Hartig and colleagues conceptualized four mechanisms by which the natural environment may improve physical and mental health: by providing clean air, spaces to be physically active, opportunities for social interactions, and the restorative, stress-reducing effects of contact with nature [8,9]. It is well established that physical activity in and of itself can reduce symptoms of depression [10,11], improve anxiety [12], and increase quality of life [13]. In several recent reviews, the availability of useable greenspace and proximity to parks were significantly associated with multiple health outcomes among urban residents, including lower weight status, reduced cardiovascular disease and diabetes risk, lower levels of anxiety and depression, reduced stress, and better quality of life [14,15,16,17,18]. Being outdoors in green spaces, away from work, and with family and friends may even provide immediate boosts in self-reported happiness, as demonstrated in a large sample of London, UK residents [2].

The associations between urban parks and mental health are likely due to a variety of factors, including connection to nature and opportunities for social interaction and physical activity. Indeed, studies suggest that social interaction in parks may increase feelings of social support and reduce feelings of loneliness [19,20]. Further, studies examining park attributes have shown that parks with a nature focus and opportunities for recreational activity are associated with positive mental health [21], while access to large green spaces may encourage physical activity participation [22,23]. Thus, the independent protective effects of greenspace and physical activity on mental health may work together to bolster their respective benefits [24,25].

Mediation analyses have been useful in studying the specific pathways through which park access improves health. For example, participation in physical activity has been shown to mediate associations between access to and time spent in greenspace and mental health outcomes such as anxiety and depression. More specifically, studies have reported physical activity mediates associations between objectively measured urban greenness and (1) depressive symptoms in pregnant women [26] and older adults [27], as well as (2) mental wellbeing in adult residents of large cities in China [28] and the Netherlands [29]. One study did not find significant mediation by nature-based active recreation, instead suggesting restorative experiences in nature may play a mediating role [30]. While evidence for a causal impact of park access on health is suggestive, research to-date has been limited by heterogeneity of wellbeing outcomes studied, lack of consensus on how to define greenspace, and use of physical activity measures that are not specific to the context of urban greenspace (e.g., park-based active recreation). In addition, few of such mediation studies have been conducted in North American cities. Therefore, the primary aim of this study was to examine whether using the park closest to home for physical activity mediated associations between the park’s proximity and past-month mental distress among adult residents of NYC. The secondary aim was to test whether inverse associations between park proximity and mental distress might be stronger among groups with higher physical activity levels, including males [31,32], dog owners [33,34], those who typically are active with other people [35], and those not concerned about neighborhood crime [36,37].

## 2. Materials and Methods

### 2.1. Study Population and Design

This study is a secondary analysis conducted in February 2020 of publicly available cross-sectional data from the Physical Activity and Transit (PAT) Survey. The NYC Department of Health and Mental Hygiene conducted the PAT Survey between 2010 and 2011. The NYC Health Department Institutional Review Board approved this study as human subjects research. Prior studies have described PAT Survey methods in detail [38,39,40]. In brief, the PAT Survey was a random-digit-dial survey (both landline and cellular phones) of adult residents of NYC aged 18 or older who were able to walk more than ten feet. The survey was designed to provide estimates of physical activity at the city, borough, and subgroup (e.g., race/ethnicity) levels. Disproportionate, equal-sized samples were collected from the five boroughs with oversampling of areas with a higher prevalence of obesity. Abt-SRBI, a survey research company based in NYC, conducted survey interviews, and the average interview length was 35 min. Of the adults who were contacted, 92% were identified as eligible and agreed to participate, resulting in an overall sample of 3811 respondents. Sample weights were applied to the data to represent the NYC non-institutionalized adult population.

### 2.2. Study Measures

All variables were derived from self-report survey measures. The mental distress outcome was based on clinical measures of general psychological distress. The variable was defined as number of days of poor mental health (i.e., “stress, depression, and problems with emotions”) during the past month (0–30 days). This survey item is a standard health-related quality of life measure utilized in public health surveillance systems [41]. Similar measures have been consistently correlated with greenness in other observational studies [42]. The park proximity exposure variable was defined as the number of minutes it would take to walk to the nearest park from home (<5 min = 1, 5–10 min = 2, 10–30 min = 3, and > 30 min = 4). We examined park proximity as a single-item continuous indicator since previous studies have found significant associations between similarly worded items and physical activity [43]. The mediator variable frequency of park use for physical activity was based on the question “how often do you use the park closest to your home for sports, exercise or other physical activity?” Response options ranged on a Likert scale from 1 = often to 4 = never. Moderator variables included gender (male or female), dog ownership by self or anyone else in household (yes or no), and whether the respondent usually participated “in activities that cause an increase in your breathing or heart rate” with another person or group, alone, or did not know or was not active in past seven days. For perceived park crime, participants were asked, “Are you concerned about crime during daylight in the park that is closest to your home?” (yes or no).

Demographic covariates included age group, gender, race/ethnicity, language spoken at home, education, employment status, annual household income, and marital status. Additional covariates were car ownership (yes or no) and perceived neighborhood retail access and traffic volume. Height and weight was utilized to calculate body mass index (BMI), and a BMI of ≥30 was categorized as obese, ≥25–29.9 as overweight, ≥20–24.9 as normal weight, and <20 as underweight [44]. To assess clinical depression, interviewers asked participants if they had ever been told by a health professional that they had depression (yes or no) [45]. Interviewers assessed self-reported moderate-to-vigorous physical activity (MVPA) using a modified version of the Global Physical Activity Questionnaire (GPAQ) [46,47]. The GPAQ categorizes meeting MVPA recommendations as accumulating at least 150 weekly minutes in bouts of at least 10 min [48].

### 2.3. Statistical Analysis

Most PAT variables on average have <2% missing values. We excluded 159 (4.2% of) observations due to missing values on one or more of the seven exposure, moderator, mediator, and outcome variables, bringing the analytic sample to 3652. We calculated descriptive statistics of demographic and health-related characteristics overall and by frequency of park use for physical activity (rarely/never vs. sometimes/often). We used a multiple regression-based approach to test for mediation, utilizing bootstrap resampling techniques (*k* = 5000) to generate 95% bias-corrected confidence intervals (BC CIs) of the indirect effect of park proximity on mental distress via park use for physical activity [49] (see Figure 1). This approach does not require a normal sampling distribution, has greater statistical power, and reduces type I errors as compared to traditional causal steps mediation methods [49].

We examined four separate mediation models in which each gender, dog ownership, usual physical activity with others, and perceived park crime were tested for significant moderation of the a-pathway, b-pathway, and c’-pathway simultaneously (see Figure 1). If a pathway was not moderated, we removed the nonsignificant interaction term from the model and repeated this process until only significant interaction terms remained. The following results are based on significant tests of moderated mediation, as indicated by at least one significant interaction term (*p*-value for interaction term <0.05) and BC CIs that did not include zero for the indirect effect and the index of moderated mediation [50]. Analysis was conducted using the PROCESS© macro version 3.3 for SAS 9.4 (SAS^®^ Institute, Inc., Cary, NC, USA).

## 3. Results

Sixty-one percent of Physical Activity and Transit (PAT) survey participants were 45 years or older, 60% were female, and 41% had at least a four-year degree. Fifty-four percent were non-white and 38% were born outside of the USA. Twenty-six percent had a BMI ≥ 30, nearly 15% had been told by a healthcare professional they had depression, and 72% reported meeting MVPA guidelines. Nearly one in four (21.6%) reported concern about crime in the park nearest their home and 15.5% of households owned a dog. Higher proportions of those who were 65 years or older, female, and black non-Hispanic reported rarely or never using the nearest park to their home for physical activity. In addition, higher proportions of those with a BMI ≥ 30, depression, not meeting MVPA guidelines, and concerns about park crime rarely or never used the park for physical activity (see Table 1).

Perceived park proximity was indirectly associated with fewer days of poor mental health via park use for physical activity, but only among those not concerned about park crime (index of moderated mediation = 0.04; SE = 0.02; 95% BC CI = 0.01, 0.09). That is, the less time individuals perceived it took to walk to the nearest park from home, the more frequently they used it to be physically active (B = 0.16, SE = 0.02, 95% CI = 0.10, 0.18). In turn, those who engaged in more frequent park-based physical activity reported fewer days of mental distress in the past month (B = −0.43, SE = 0.12, 95% CI = −0.65, −0.20). Indirect associations did not depend on gender, dog ownership, or usual physical activity with others (*p*-value for interaction term ≥ 0.05) (see Figure 2).

## 4. Discussion

Using findings from a population-representative study of NYC residents with active recreation data, we found that engaging more frequently in physical activity in a park near home may explain why urban-dwelling adults living closer to a park experience fewer days of mental distress. However, the benefits of living near greenspace may depend on park conditions such as adequate safety. This study expands prior neighborhood and mental health research with a mechanistic examination of physical activity that takes place in park settings.

These findings corroborate the limited prior research in USA cities, in which overall physical activity partially mediated associations between residential greenness assessed using a Normalized Difference Vegetation Index (NDVI) and (1) perceived stress among older adults [27] and (2) depressive symptoms in pregnant women [26]. A few recent studies have examined physical activity in combination with social factors as mediators of associations between greenspace and mental health. Loneliness [20,51], and social cohesion [28,29,51] were significant, and in some cases stronger mediators than physical activity of the associations between neighborhood resources for physical activity (greenspace and streetscape greenery) and wellbeing and depressive symptoms outcomes. One study did not demonstrate significant indirect associations between residential proximity to neighborhood greenspace and mental wellbeing via leisure-time physical activity, social cohesion, nor perceived stress; however, neighborhood satisfaction significantly mediated the association [9]. Future studies should also capture multiple physical environmental, social, behavioral and stress-reducing/restorative mechanisms to understand their potentially synergistic contributions to the mental health-promoting influence of neighborhood greenness.

Concerningly, we also identified that the benefits of living near greenspace may not be equitably experienced, with benefits depending on perceived park conditions such as adequate safety. In general, evidence has been inconsistent that safety concerns modify positive associations between neighborhood environments and physical activity [52]. In this study, the significant moderation of the *a* pathway between park proximity and park use for physical activity suggests that perceptions of park access and park crime interact, explaining more frequent physical activity in the park—only among those not expressing safety concerns—which in turn may promote mental health—only in safer (and likely more affluent) neighborhoods. Indeed, among 2775 adult residents of Melbourne, Australia, “personally feeling safe” going to the park ranked higher in importance for encouraging park-based physical activity than the park being “easy to get to” [37]. In a survey of 3815 USA adults living within one kilometer of an urban park, those who perceived the park as safe (88%) had a 4.6 times greater odds of having visited the park [53], whereas higher objectively-measured violent crime was associated with significant reductions in both park use and park-based physical activity in low-income urban neighborhoods [54,55]. Prior studies of socioeconomically disadvantaged neighborhoods suggest that greater exposure to social stressors like crime may offset the physical activity and mental health benefits of walkability [56,57] and park access [57,58,59]. However, improving walkability, incivilities, and aesthetics surrounding parks in low-income areas increases visits to underutilized greenspace [60]. Addressing neighborhood safety concerns also may help to ease the higher burden of depression among residents of deprived areas who report that where they live is not safe from crime [6,61,62].

When promoting physical activity within parks, public health and urban planning professionals may find it necessary to improve both perceived and material safety in the community in order to maximize the mental health benefits of nearby parks. Several park features may foster perceptions of park safety, including reducing vegetation density and installing street lights along trails [63], addressing park cleanliness and incivilities [37], and offering organized activities [53,64]. Community interventions that promote racial inclusion and improve race relations also may help Black and Latino individuals to feel comfortable and accepted in park settings [65]. Male gender, dog ownership, or usual physical activity with others did not moderate direct or indirect associations, suggesting that provided one perceives nearby parks as safe, the mental health benefits of regular park-based physical activity likely are equally available to females (who are 51% of all park users) and those who do not own a dog, or who otherwise prefer to be active alone [66].

When research elucidates viable mechanisms, health-promoting nature-based interventions can be designed and tested. Recent evidence syntheses of nature-based interventions to promote physical activity [67,68] and mental health [69] suggest effectiveness. Currently underway are a few such interventions that integrate primary care providers. In these programs, a physician’s recommendation to visit parks to be active and experience nature are combined with intervention components like behavioral counseling and environmental education [70,71,72]. In addition, low-cost park-based physical activity interventions hold promise for augmenting or expanding the reach of mental health promotion beyond traditional community mental health services, particularly in low-income communities where access to mental healthcare for psychiatric medications and psychotherapy may be limited. Even with accessible mental health care, patients cite the stigma associated with seeking care to be a significant barrier [73], whereas communities may more readily accept effective nature-based physical activity interventions.

This study’s limitations include a cross-sectional design which precludes causal inference, and self-reported survey measures which may introduce social desirability and same-source biases. The mental distress measure, which was not a validated, multifactorial questionnaire nor a structured diagnostic interview, limited this study’s outcome variable to the self-assessed absence of mental illness, rather than the presence of emotional wellbeing. Perceived access and proximity to urban parks, neighborhood greenness, and streetscape greenery are related constructs; however, they differ conceptually from one another and from geographic information systems (GIS)-based measures [74,75]. Consensus also is lacking on definitions of greenness [15], which poses challenges to a meaningful synthesis of the available evidence. Mental distress observations may cluster geographically, and our inability to account for neighborhood-level variables such as area socioeconomic deprivation, which may explain geographic variability in mental distress, is an important limitation of this study. While both perceived safety from crime and objectively-measured crime rates are associated with physical activity [76], perceptions of neighborhood safety and crime differ conceptually from objective neighborhood crime data. Perceived crime measures may reflect personal histories with crime, as well as cognitive, emotional and behavioral responses to crime, thus conceptual distinctions among safety and crime measures should be clearly delineated in future research [77]. Since perceived park crime may be a proxy for neighborhood socioeconomic status, we attempted to mitigate potential confounding to the extent possible by including individual annual household income from all sources as percent of Federal Poverty Level in the statistical models. Current conditions of individual NYC neighborhoods likely differ from when PAT data were collected in 2010–2011, albeit environmental change tends to be slow. Future parks research should utilize hierarchical or multilevel modeling techniques to examine associations among neighborhood socioeconomic status, objectively-measured greenness and crime, environmental perceptions, and health behaviors and outcomes.

Strengths of this study include the fact that the PAT survey is a multilingual survey administered to randomly selected NYC residents with a high rate of cooperation (92%). Survey data were weighted to adjust for the probability of selection and differential nonresponse. Results of this study may not generalize to institutionalized adults, those living in college dormitories, or those who could not be reached by landline or mobile phone. Additionally, NYC residents are half as likely to be physically inactive as adults nationwide [40,78] and 76.5% live within ¼ mile of a park [79]. For individuals living less than ¼ mile from a park, associations between park proximity and mental health may be stronger [80]. Thus, results of this study may be particularly relevant to other economically developed, high-density, walkable cities in which parks are within walking distance of the majority of residents’ homes. In general, urban residents of developed countries in the Global North tend to have more equitable park proximity than urban residents of developing countries of the Global South; however, socioeconomic inequities in park quantity and quality (which may be particularly relevant to mental health) persist in cities throughout the world [81,82]. This study’s findings contribute to an international conversation about ways in which the natural environment can be protected and leveraged through actionable policy change to equitably improve physical and mental health in an increasingly urbanized world [83].

## 5. Conclusions

Closer park proximity was significantly indirectly associated with less past-month mental distress via neighborhood park use for physical activity among urban residents who were not concerned about park crime as compared to those concerned about park crime. Male gender, dog ownership, or usual physical activity with others did not moderate direct or indirect associations. In order to maximize the mental health benefits of nearby parks, future research should design and test park-based physical activity interventions and assess the impact of park features that improve perceptions of park safety, promote physical activity, and subsequently improve mental health and wellbeing outcomes, particularly in socioeconomically disadvantaged neighborhoods.

## Figures and Tables

**Figure 1 ijerph-17-04885-f001:**
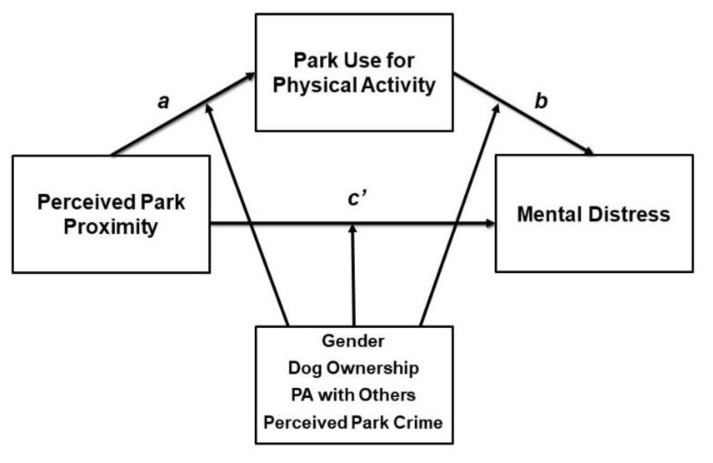
Conceptual Model of Park Proximity on Mental Distress via Park Use for Physical Activity (PA). *a* depicts the association between perceived park proximity and park use for PA, *b* depicts the association between park use for PA and mental distress, and *c′* depicts the (direct) association between perceived park proximity and mental distress, controlling for park use for PA.

**Figure 2 ijerph-17-04885-f002:**
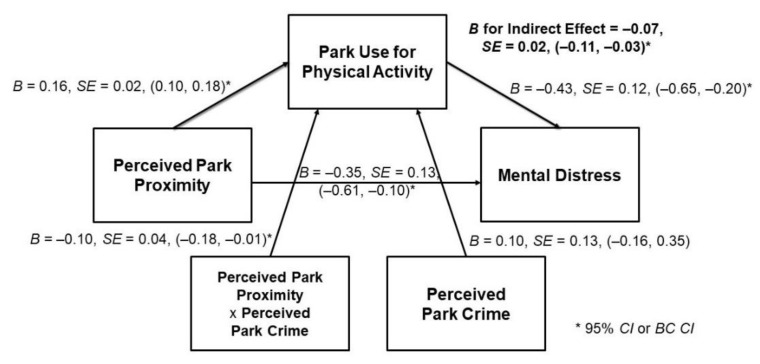
Direct and indirect associations between park proximity and mental distress among 3652 New York City residents. Model adjusted for age group, gender, race, language of interview, education, marital status, employment status, median household income, body mass index, car ownership, perceived traffic volume, perceived retail access, survey wave and survey strata. Abbreviations: *B* is beta coefficient, *SE* is standard error, *CI* is confidence interval, *BC CI* is bias-corrected confidence interval, and x signifies the multiplication of two variable means to yield an interaction term. * Indicates a *CI* or *BC CI* is statistically significant based on a probability of 95%.

**Table 1 ijerph-17-04885-t001:** Characteristics of New York City Physical Activity (PA) & Transit Survey Participants.

Categorical Variables	All Participants	Sometimes/Often Park-Based PA	Rarely/Never Park-Based PA
	*N* = 3652	*N* = 1510	*N* = 2142
	*N* (%)	*N* (%)	*N* (%)
Age group			
18–24 years	266 (7.3)	114 (7.5)	152 (7.1)
25–44 years	1139 (31.2)	553 (36.6)	586 (27.4)
45–64 years	1400 (38.3)	567 (37.5)	833 (38.9)
65+ years	839 (23.0)	273 (18.1)	566 (26.4)
Missing	8 (0.2)	3 (0.2)	5 (0.2)
Gender			
Female	2151 (58.9)	839 (55.6)	1312 (61.3)
Male	1501 (41.1)	671 (44.4)	830 (38.7)
Race			
White, non-Hispanic	1598 (43.8)	673 (44.6)	925 (43.2)
Black, non-Hispanic	867 (23.7)	318 (21.1)	549 (25.6)
Hispanic	819 (22.4)	364 (24.1)	455 (21.2)
Asian or Pacific Islander	293 (8.0)	124 (8.2)	169 (7.9)
Other ^a^	75 (2.1)	31 (2.1)	44 (2.1)
Born in United States			
Yes	2253 (61.8)	921 (61.0)	1332 (62.2)
Language spoken at home			
English	2818 (77.2)	1148 (76.0)	1670 (78.0)
Spanish	457 (12.5)	218 (14.4)	239 (11.2)
Russian	95 (2.6)	28 (1.9)	67 (3.1)
Chinese	131 (3.6)	42 (2.8)	89 (4.2)
Other	139 (3.8)	70 (4.6)	69 (3.2)
Missing	12 (0.3)	4 (0.3)	8 (0.4)
Employment status			
Employed	1976 (54.1)	855 (56.6)	1121 (52.3)
Unemployed	325 (8.9)	151 (10.0)	174 (8.1)
Not in the labor force	1340 (36.7)	498 (33.0)	842 (39.3)
Missing	11 (0.3)	6 (0.4)	5 (0.2)
Education			
Some high school	463 (12.7)	189 (12.5)	274 (12.8)
High school graduate or equivalent	904 (24.8)	350 (23.2)	554 (25.9)
Some college	778 (21.3)	302 (20.0)	476 (22.2)
4-year college graduate or higher	1496 (41.0)	669 (44.3)	827 (38.6)
Missing	11 (0.3)	0 (0.0)	11 (0.5)
Marital status			
Married or living together	1607 (44.0)	733 (48.5)	874 (40.8)
Divorced or separated	644 (17.6)	253 (16.8)	391 (18.3)
Widowed	378 (10.4)	115 (7.6)	263 (12.3)
Never married	989 (27.1)	401 (26.6)	588 (27.5)
Missing	34 (0.9)	8 (0.5)	26 (1.2)
Tobacco use (past 30 days)			
Most or all days	377 (10.3)	135 (8.9)	242 (11.3)
Some days	194 (5.3)	87 (5.8)	107 (5.0)
Never	3078 (84.3)	1287 (85.2)	1791 (83.6)
Missing	3 (0.1)	1 (0.1)	2 (0.1)
Physical activity level ^b^			
Sufficiently active	2639 (72.3)	1196 (79.2)	1443 (67.4)
Insufficiently active	692 (18.9)	238 (15.8)	454 (21.2)
Inactive	316 (8.7)	75 (5.0)	241 (11.3)
Missing	5 (0.1)	0 (0.0)	4 (0.2)
Usually physically active with others			
Yes, with another person or group	746 (20.4)	397 (26.3)	349 (16.3)
No, alone	1469 (40.2)	603 (39.9)	866 (40.4)
Inactive in past 7 days or Don’t know	1437 (39.4)	510 (33.8)	927 (43.3)
Body mass index (BMI)			
Underweight	72 (2.0)	23 (1.5)	49 (2.3)
Normal weight	1353 (37.0)	620 (41.1)	733 (34.2)
Overweight	1250 (34.2)	509 (33.7)	741 (34.6)
Obese	955 (26.2)	350 (23.2)	605 (28.2)
Missing	22 (0.6)	8 (0.5)	14 (0.7)
History of depression diagnosis			
Yes	536 (14.7)	196 (13.0)	340 (15.9)
No	3108 (85.1)	1313 (87.0)	1795 (83.8)
Missing	8 (0.2)	1 (0.1)	7 (0.3)
Activity limitations due to health			
Yes	750 (20.5)	240 (15.9)	510 (23.8)
Concern about park crime during day			
Yes	788 (21.6)	284 (18.8)	504 (23.5)
No or Don’t know	2864 (78.4)	1226 (81.2)	1638 (76.5)
Car ownership			
Yes	1617 (44.3)	674 (44.6)	943 (44.0)
No	2024 (55.4)	831 (55.0)	1193 (55.7)
Missing	11 (0.3)	5 (0.3)	6 (0.3)
Dog ownership			
Yes	566 (15.5)	279 (18.5)	287 (13.4)
** Continuous Variables**	***M* (*SD*)**	***M* (*SD*)**	***M* (*SD*)**
Mental distress (days in past 30)	3.7 (7.7)	3.1 (6.8)	4.2 (8.3)
Park use for physical activity ^c^	2.2 (1.1)	3.4 (0.5)	1.4 (0.5)
Park proximity to home ^d^	2.7 (1.0)	2.9 (0.9)	2.6 (1.0)

^a^ Other race includes American Indian, Alaska Native, mixed race, and other race. ^b^ Physical activity level based on meeting 2008 recommendations for moderate-vigorous physical activity. ^c^ Average frequency of park use for physical activity approximates “rarely” among options “often”, “sometimes”, “rarely” and “never”. ^d^ Average proximity of nearest park to home reflects about 10 min by walking. Abbreviations: PA is physical activity, *N* is number, *M* is mean, and *SD* is standard deviation.

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
