# Peer review of "Park Proximity and Use for Physical Activity among Urban Residents: Associations with Mental Health"

_ijerph, 2020, doi:10.3390/ijerph17134885_

Round 1

Reviewer 1 Report

This paper is a tightly focused and carefully argued look at an important issue. It provides a thorough overview of the literature it is addressing, and makes a significant contribution to it. It should be published.

My one comment is that in the discussion, and perhaps in future research, the authors might want to more carefully distinguish safety and crime from people *perceptions* of safety and threat. Especially in urban areas there are good reasons to think that people’s perceptions of neighborhoods, and the safety and crime rates of those neighborhoods, can decouple from objective boundaries and the actual safety and crime rates of a neighborhood.

Author Response

Reviewer 1 - Comments and Suggestions for Authors

This paper is a tightly focused and carefully argued look at an important issue. It provides a thorough overview of the literature it is addressing, and makes a significant contribution to it. It should be published.

We appreciate the Reviewer’s interest and encouraging comments.

My one comment is that in the discussion, and perhaps in future research, the authors might want to more carefully distinguish safety and crime from people *perceptions* of safety and threat. Especially in urban areas there are good reasons to think that people’s perceptions of neighborhoods, and the safety and crime rates of those neighborhoods, can decouple from objective boundaries and the actual safety and crime rates of a neighborhood.

We agree with the Reviewer that clearly delineating perceived and objective crime measures is critical, and should be suggested for future research. We’ve taken the following steps to address the Reviewer’s concerns:

1) In the third paragraph of the discussion section, at line 202, 206, and 212, we have indicated whether the mentioned safety or crime measures were perceived or objective measures if not previously specified.

2) We have added the Reviewer’s important point as part of the discussion. At line 256, we have added, “While both perceived safety from crime and objectively-measured crime rates are associated with physical activity [76], perceptions of neighborhood safety and crime differ conceptually from objective neighborhood crime data. Perceived crime measures may reflect personal histories with crime, as well as cognitive, emotional and behavioral responses to crime, thus conceptual distinctions among safety and crime measures should be clearly delineated in future research [77].”

Rees-Punia, E.; Hathaway, E.D.; Gay, J.L. Crime, perceived safety, and physical activity: A meta-analysis. Prev Med 2018, 111, 307-313, doi:10.1016/j.ypmed.2017.11.017.

Sallis, J.F. CRIME-PA: CRIme Measures Evaluations for Physical Activity. National Heart, Lung, and Blood Institute (NHLBI), National Institute of Health (NIH): University of California San Diego, 2016.

Reviewer 2 Report

Thank you for an interesting paper. Overall, I found the paper to be well written and well presented. My main concern is that the title of the paper is not well reflected in the content. I was expecting to be reading about the role of perceived crime and mental health, however this paper is a more broad, general view of the mediating factors around park-based physical activity. Therefore, my specific suggestions below:

  1. Suggest a change in title to reflect the general nature of the paper (more than 1 mediating factor) or add further content to emphasise the role of perceived crime, particularly in the introduction.
  2. The abstract could be improved. The first line sounds like a finding rather than background information. The concluding sentence is good but does not reflect the title or provide a strong case regarding the moderating role of perceived crime with mental health (the sentence before this might be a better conclusion). 
  3. Line 39: What is the relationship between obesity and contact with the natural environment? This sentence seems out of place without context.  
  4. The methods are well described, although it may be beneficial to define "mental distress". 
  5. The results are clearly presented.
  6. The discussion is well written and reflects the findings well. It would be beneficial to include the dates of the survey in the limitations as this data is almost 10 years old. Your future research suggestions are well aligned with your study. I look forward to reading it. 

Author Response

Reviewer 2 - Comments and Suggestions for Authors

Thank you for an interesting paper. Overall, I found the paper to be well written and well presented. My main concern is that the title of the paper is not well reflected in the content. I was expecting to be reading about the role of perceived crime and mental health, however this paper is a more broad, general view of the mediating factors around park-based physical activity. Therefore, my specific suggestions below:

We thank Reviewer 2 for their interest and thoughtful suggestions for improving the paper.

  1. Suggest a change in title to reflect the general nature of the paper (more than 1 mediating factor) or add further content to emphasise the role of perceived crime, particularly in the introduction.

Thank you for this suggestion. We agree that the title over-emphasized park crime. At lines 3 and 4, we deleted “the moderating role of perceived crime” from the title so that the title is both shorter and more reflective of the entire manuscript. The title is now “Park Proximity and Use for Physical Activity Among Urban Residents: Associations with Mental Health”.

  1. The abstract could be improved. The first line sounds like a finding rather than background information. The concluding sentence is good but does not reflect the title or provide a strong case regarding the moderating role of perceived crime with mental health (the sentence before this might be a better conclusion). 

Thank you for these suggestions to improve the abstract. We changed the first few sentences of the abstract to better introduce the problem. The first few sentences of the abstract starting at line 17 now read, “Increasing global urbanization limits interaction between people and natural environments, which may negatively impact population health and wellbeing. Urban residents who live near parks report better mental health. Physical activity (PA) reduces depression and improves quality of life. Despite PA’s protective effects on mental health, the added benefit of urban park use for PA is unclear.”

We also added the word “safety” to the concluding paragraph to provide more of the nuance of our study. The concluding sentence at line 32 now reads, “Investment in park safety and park-based PA promotion in urban neighborhoods may help to maximize the mental health benefits of nearby parks.”

  1. Line 39: What is the relationship between obesity and contact with the natural environment? This sentence seems out of place without context.  

We apologize for the confusion. Obesity was intended as one example of the many physical health conditions with epidemiological associations to greenspace, most likely through increased physical activity. At line 42, we have edited the text to clarify obesity as an example. At line 53, we also cited an additional systematic review explaining the link between greenspace and obesity.

Lachowycz, K.; Jones, A.P. Greenspace and obesity: a systematic review of the evidence. Obesity reviews 2011, 12, e183-e189.

  1. The methods are well described, although it may be beneficial to define "mental distress". 

Thank you. We expanded upon the mental distress construct. Starting at line 101, we also further outlined the mental distress measure in response to suggestions by Reviewer 3. The section now reads, “The mental distress outcome was based on clinical measures of general psychological distress. The variable was defined as number of days of poor mental health (i.e., “stress, depression, and problems with emotions”) during the past month (0-30 days). This survey item is a standard health-related quality of life measure utilized in public health surveillance systems [41]. Similar measures have been consistently correlated with greenness in other observational studies [42].”

James, P.; Banay, R.F.; Hart, J.E.; Laden, F. A Review of the Health Benefits of Greenness. Curr Epidemiol Rep 2015, 2, 131-142, doi:10.1007/s40471-015-0043-7.

  1. The results are clearly presented.

Thank you.

  1. The discussion is well written and reflects the findings well. It would be beneficial to include the dates of the survey in the limitations as this data is almost 10 years old. Your future research suggestions are well aligned with your study. I look forward to reading it. 

Thank you! To address the Reviewer’s concern about the age of the data, we have added the following sentence to the limitations paragraph at line 263: “Current conditions of individual NYC neighborhoods likely differ from when PAT data were collected in 2010-2011, albeit environmental change tends to be slow.”

Reviewer 3 Report

First of all I very much enjoyed reading this article.  It was comprehensive, logical and very clearly written; the methods and statistics were appropriate and well justified and the conclusions followed logically from the article itself. The mediation analyses are well conducted and well reported and the authors have created some strong models from what could potentially have been a confusing picture given the number of variables available for analysis.  Overall a really strong submission.  I have however a couple of suggestions that would potentially strengthen this further or that the authors may wish to consider.

Suggestions for strengthening the existing submission:

1) I felt in places - both in the introduction and discussion sections that the research was a little US centric. This was despite cross cultural comparisons being drawn in some of the text; I felt that this was insufficient and would instead like to see at least in the discussion if there are "take home messages" or "recommendations for practice" that could apply beyond the USA.  This would not take much additional writing but might take a little bit of additional thinking.

2) Given the focus on social, urban and natural environments in this year's World Happiness Report, I was really surprised not to see connections being made to this.  There are some interesting trends in terms of well-being and connections to urban and rural environments which seem to reflect Western/Non-Western country differences.  It would be worth including consideration of this in the current study.

3) Although the authors are constrained by the secondary nature of the data set, I was surprised to see no reflection on the overall appropriateness of the mental distress measure.  While it would of course not be possible to use other measures and this does have some strong elements (e.g. focus on last 30 days and on multiple indices of mental distress), some critical appraisal of the suitability of this tool would be useful.  It is almost as though the authors have accepted the position that this is a standard measure used by national agencies and therefore this makes it a suitable tool.  This may be the case and there are similarities between the authors findings and other studies using different measures of well-being.  However the assumption that this is "automatically good because of its prevalence" comes out from the writing and contributes to the feel of this article as being a little culturally constricted.

I should emphasise that I do not see any of these as significant issues and each could be addressed in a short and straightforward way.  However these would potentially widen the appeal of the research for an international audience in particular; especially given the international scope of this journal.

Author Response

Reviewer 3 - Comments and Suggestions for Authors

First of all I very much enjoyed reading this article.  It was comprehensive, logical and very clearly written; the methods and statistics were appropriate and well justified and the conclusions followed logically from the article itself. The mediation analyses are well conducted and well reported and the authors have created some strong models from what could potentially have been a confusing picture given the number of variables available for analysis.  Overall a really strong submission.  I have however a couple of suggestions that would potentially strengthen this further or that the authors may wish to consider.

We really appreciate the Reviewer’s enthusiasm for the article and their encouraging comments. We feel that incorporating the Reviewer’s suggestions has strengthened the manuscript. We outline the changes below.

Suggestions for strengthening the existing submission:

1) I felt in places - both in the introduction and discussion sections that the research was a little US centric. This was despite cross cultural comparisons being drawn in some of the text; I felt that this was insufficient and would instead like to see at least in the discussion if there are "take home messages" or "recommendations for practice" that could apply beyond the USA.  This would not take much additional writing but might take a little bit of additional thinking.

We feel the Reviewer has made a fair assessment of the focus on the U.S. and North America. In several instances, we have sought to balance our discussion by incorporating more of the state of the science worldwide.

a) In the introduction we added, at line 38, the prevalence of urban living worldwide. At line 43, we cited global statistics demonstrating increases in depressive disorders worldwide. The first few sentences of the Introduction now read, “Today, the world population is increasingly concentrated in cities. In 2018, 82% of North Americans and 55% of people worldwide lived in urban areas [1]. Living in large cities limits regular contact with the natural environment and the associated physical and mental health benefits. Lack of quality greenspace may exacerbate downward trends in urban residents’ subjective wellbeing, particularly those of low-income, in the world’s most economically developed cities [2]. The prevalence of chronic diseases (e.g., obesity) and depression have increased significantly in the United States (US) and globally in recent decades [3-5]…”

James, S.L.; Abate, D.; Abate, K.H.; Abay, S.M.; Abbafati, C.; Abbasi, N.; Abbastabar, H.; Abd-Allah, F.; Abdela, J.; Abdelalim, A., et al. Global, regional, and national incidence, prevalence, and years lived with disability for 354 diseases and injuries for 195 countries and territories, 1990–2017: a systematic analysis for the Global Burden of Disease Study 2017. The Lancet 2018, 392, 1789-1858, doi:10.1016/s0140-6736(18)32279-7.

Helliwell, J.F.; Layard, R.; Sachs, J.; De Neve, J.-E., eds. World Happiness Report 2020; Sustainable Development Solutions Network: New York, NY, 2020.

b) In the concluding paragraph, we have highlighted findings from a review suggesting uneven access to high-quality parks in countries across the globe, and that quality may have particular relevance from a mental health perspective. Starting at line 276, the last 3 sentences of the paper now read, “results of this study may be particularly relevant to other economically developed, high-density, walkable cities in which parks are within walking distance of the majority of residents’ homes. In general, urban residents of developed countries in the Global North tend to have more equitable park proximity than urban residents of developing countries of the Global South; however, socioeconomic inequities in park quantity and quality (which may be particularly relevant to mental health) persist in cities throughout the world [81,82]. This study’s findings contribute to an international conversation about ways in which the natural environment can be protected and leveraged through actionable policy change to equitably improve physical and mental health in an increasingly urbanized world [83].”

Rigolon, A.; Browning, M.; Lee, K.; Shin, S. Access to Urban Green Space in Cities of the Global South: A Systematic Literature Review. Urban Science 2018, 2, doi:10.3390/urbansci2030067.

Francis, J.; Wood, L.J.; Knuiman, M.; Giles-Corti, B. Quality or quantity? Exploring the relationship between Public Open Space attributes and mental health in Perth, Western Australia. Soc Sci Med 2012, 74, 1570-1577, doi:10.1016/j.socscimed.2012.01.032.

Salgado, M.; Madureira, J.; Mendes, A.S.; Torres, A.; Teixeira, J.P.; Oliveira, M.D. Environmental determinants of population health in urban settings. A systematic review. BMC Public Health 2020, 20, 853, doi:10.1186/s12889-020-08905-0.

2) Given the focus on social, urban and natural environments in this year's World Happiness Report, I was really surprised not to see connections being made to this.  There are some interesting trends in terms of well-being and connections to urban and rural environments which seem to reflect Western/Non-Western country differences.  It would be worth including consideration of this in the current study.

Thank you for bringing relevant results of the World Happiness Report to our attention. It has provided a nuanced understanding of wellbeing in the world’s most economically developed cities. We have added the following sentence to the Introduction, starting at line 40. “Lack of quality greenspace may exacerbate downward trends in urban residents’ subjective wellbeing, particularly those of low-income, in the world’s most economically developed cities [2].”

It also has provided a nice addition of the more immediate effects of spending time in parks on happiness, starting at line 53. “Being outdoors in green spaces may even provide immediate boosts in self-reported happiness, as demonstrated in a large sample of London, UK residents [2].”

Helliwell, J.F.; Layard, R.; Sachs, J.; De Neve, J.-E., eds. World Happiness Report 2020; Sustainable Development Solutions Network: New York, NY, 2020.

3) Although the authors are constrained by the secondary nature of the data set, I was surprised to see no reflection on the overall appropriateness of the mental distress measure.  While it would of course not be possible to use other measures and this does have some strong elements (e.g. focus on last 30 days and on multiple indices of mental distress), some critical appraisal of the suitability of this tool would be useful.  It is almost as though the authors have accepted the position that this is a standard measure used by national agencies and therefore this makes it a suitable tool.  This may be the case and there are similarities between the authors findings and other studies using different measures of well-being.  However the assumption that this is "automatically good because of its prevalence" comes out from the writing and contributes to the feel of this article as being a little culturally constricted.

The Reviewer makes a great point regarding the suitability of the mental distress measure. It was not our intention to convey that we deemed it the most suitable measure. We have adjusted the writing to focus on similarities between our findings and the findings of other studies rather than the prevalence of its use in the U.S. The first part of the first paragraph of the Study Measures section, starting at line 100, now reads, “All variables were derived from self-report survey measures. The mental distress outcome was based on clinical measures of general psychological distress. The variable was defined as number of days of poor mental health (i.e., “stress, depression, and problems with emotions”) during the past month (0-30 days). This survey item is a standard health-related quality of life measure utilized in public health surveillance systems [41]. Similar measures have been consistently correlated with greenness in other observational studies [42].

James, P.; Banay, R.F.; Hart, J.E.; Laden, F. A Review of the Health Benefits of Greenness. Curr Epidemiol Rep 2015, 2, 131-142, doi:10.1007/s40471-015-0043-7.

We would also like to refer the Reviewer to our slightly expanded discussion of the methodological limitations of the mental distress measure starting at line 247. This sentence now reads, “The mental distress measure, which was not a validated, multifactorial questionnaire nor a structured diagnostic interview, limited this study’s outcome variable to the self-assessed absence of mental illness, rather than the presence of emotional wellbeing.”

I should emphasise that I do not see any of these as significant issues and each could be addressed in a short and straightforward way.  However these would potentially widen the appeal of the research for an international audience in particular; especially given the international scope of this journal.

Thank you, we appreciate your consideration of the scientific community in its broadest (global) sense.